# Identifying "Candidates for University Projects" in Open Source Software

## Abstract

Open Source Software (OSS) and its accompanying communities are a valuable environment for university students to gain real-world collaborative development experience. At the same time, tasks are often too complex and poorly scoped, making it challenging to identify tasks suitable for students looking to contribute to these communities. Our goal is to automate the discovery of OSS issues that are both educationally valuable and technically feasible for university-level coursework. We compare the performance of a supervised Machine Learning (ML) classifier and a Large Language Model (LLM) baseline, both trained on a labeled dataset of GitHub issues, for flagging issues as *"Candidates for University Projects"* and identifying their characteristics. Despite extreme class imbalance (1.6% positive rate), the Random Forest classifier was able to identify ~45% (5 out of 11) of the JabRef "Candidate University Projects" (out of 30 issues recommended) tagged in the repository. The project maintainer reviewed the top-30 issues identified by the classifier and identified another 13 candidates, bringing the total to 18 (with k=30; 60% precision). This demonstrates the model's practical utility in supporting human triage. The LLM baseline achieved low recall and precision, with limited effectiveness compared to the supervised learning approach. Our study provides insights for educators, students, and OSS maintainers seeking to streamline the identification of academic project tasks. It also suggests that lightweight models can uncover valuable tasks even in noisy, under-annotated repositories, pointing toward a scalable triage process.

## Keywords

Open Source Software, Issue Classification, Random Forest

**ACM Reference Format:**
Anonymous Author(s). 2026. Identifying "Candidates for University Projects" in Open Source Software. In . ACM, New York, NY, USA, 9 pages. https://doi.org/10.1145/nnnnnnn.nnnnnnn

## 1 Introduction

Open Source Software (OSS) has become a powerful environment for learning software engineering, offering real-world, collaborative environments where students can apply technical skills and contribute to active projects [20]. As academic programs increasingly integrate OSS into coursework, newcomers often struggle to find tasks that match their skills and can be completed within a course timeline, while project maintainers are frequently too time-constrained to curate, scope, and annotate issues to make suitable starter tasks easy to identify. [8, 22]. Prior studies on OSS communities have also emphasized that mentoring and Q&A support for newcomers is a recurring burden in OSS communities, motivating conversational support tools and the need for evaluation datasets tailored to newcomer mentoring [6]. OSS repositories usually do not explicitly tag tasks as suitable for academic engagement, making the discovery of educationally viable issues a manual process. Although labels such as Good First Issue help new contributors start contributing to an OSS project [24], they are typically scoped for one-time, low-effort contributions, instead of meaningful opportunities for academic projects.

The difficulty in triaging or creating these projects for university students arises from both the scale and the unstructured nature of OSS issue trackers. Projects may contain hundreds or thousands of open issues, varying widely in scope, complexity, and suitability [3, 7]. While some tasks involve simple bug fixes or documentation improvements, others require a deep understanding of architecture or long-term coordination [14]. For instructors seeking to align academic goals with project needs, this lack of structure creates friction, while students often risk selecting issues that exceed their knowledge or available time [20].

In this work, we study this challenge: the unscalable and manual nature of identifying pedagogically appropriate tasks in open-source issue trackers for academic use, in the context of JabRef [13], an OSS reference manager primarily written in Java that supports academic writing and bibliography management. JabRef maintainers already curate a list of "Candidates for University Projects" to identify these tasks [12]. However, this manual process is time-consuming and does not scale. Automating the identification of OSS tasks, appropriate for university projects, would reduce this burden, improve matching quality, and enable more sustainable collaborations between academia and open source communities [29]. In this context, this paper aims to evaluate automated approaches for identifying and characterizing OSS issues that are pedagogically appropriate for academic use (Candidates for University Projects). To this end, we explore the following research questions:

- **RQ1:** To what extent can OSS issues be automatically classified as *Candidates for University Projects*?
- **RQ2:** To what extent can we predict relevant contextual characteristics of Candidate issues?

To address these research questions, we built a classification pipeline that examines the information provided in each issue (title, body, and image descriptions) to determine its suitability for university projects. We trained a Random Forest model on the dataset labeled by project maintainers, addressing class imbalance through stratified data splits and positive-only augmentation, and evaluating the model using a ranking-based Top-$K$ selector to directly control the precision–recall trade-off in a setting where positive

cases are rare. We also compare the model's predictions with those of a Large Language Model, assessing how well machine-learned and language-based reasoning align in this task. To follow up, we also presented these predictions to the project maintainer, who further identified the correctness of the outcomes.

Our results show that despite limited data and class imbalance, the Random Forest classifier retrieved plausible candidates not marked by maintainers. When shared with JabRef's maintainer, more than half of the new suggestions were confirmed as suitable for university projects, demonstrating the model's ability to recommend valuable tasks overlooked in manual triage. This highlights the practical benefit of machine learning in expanding the pool of university candidate issues. The few-shot LLM achieved low recall and low precision, demonstrating limited effectiveness compared to the Random Forest model. Random Forest significantly outperforms the LLM approach, providing both better ranking quality and more reliable identification of candidate issues. A rank-then-review workflow can streamline instructor effort by turning large issue sets into a manageable weekly shortlist for academic use.

## 2 Background and Related Work

Open Source Software provides a dynamic platform for computer science education, bridging academic instruction and professional practice [10]. Because most OSS projects are maintained publicly and actively welcome community contributions, students can participate directly in real-world software development tasks. These engagements foster technical proficiency and such as essential soft skills like collaboration and communication [20]. Educators who integrate OSS into their curricula report enhanced student learning outcomes, particularly when students contribute to live codebases rather than work on isolated, toy projects [22]. Formal initiatives such as Google Summer of Code (GSoC) exemplify successful academia-focused initiatives, with students mentored by OSS maintainers while contributing to established repositories [8, 23].

*Newcomer Task Identification in OSS Projects* A key obstacle for students and newcomer contributors is the difficulty of identifying suitable first issues. Projects often label tasks as "Good First Issue" (GFI) or "help wanted," but research shows that such tags are inconsistently applied and sometimes misleading [24]. Studies have revealed that experienced developers frequently resolve issues tagged as GFI, indicating a misalignment between the label intent and the actual task complexity [24]. To improve onboarding, maintainers use manual strategies like scaffolding tasks and documenting issue contexts, but these practices are time-intensive and not scalable [3]. Automated solutions have emerged to address this gap. For example, RECGFI employs supervised learning to predict newcomer-friendly issues, achieving notable accuracy in identifying suitable tasks even when no GFI label exists [29]. These efforts highlight the growing interest in automating issue triage for onboarding.

*Machine Learning for Issue Classification* Machine learning (ML) has long been used to classify software issues, distinguishing between bugs, features, and support queries. Misclassification in bug tracking systems can lead to flawed defect prediction models, emphasizing the importance of accurate issue labeling [14]. Subsequent work applied Naive Bayes, SVM, and logistic regression to categorize tickets based on textual content [15]. Recently, more sophisticated models like Random Forests and gradient boosting have been used for severity and priority prediction [7]. In educational contexts, ML-based classifiers can filter out tasks unsuitable for students and identify issues appropriate for coursework or capstone projects.

*LLMs in Software Engineering* The advent of large language models (LLMs) such as BERT, Codex, and GPT-4 has transformed software engineering automation. These models demonstrate remarkable capabilities in code generation, summarization, and natural language understanding [19, 30].

Prior work on in-context learning shows that adding a small number of demonstrations can improve language-model text classification performance compared to zero-shot prompting, and can sometimes be competitive with fine-tuned baselines, although results can be sensitive to prompt format and example selection or ordering [4, 32]. Developers increasingly use LLMs in practice, consulting ChatGPT for explanations, bug fixes, and design discussions [16]. This suggests that LLMs can complement traditional ML techniques in issue classification and triage.

Our study contributes a complementary angle on newcomer support by targeting educational fit rather than general newcomer friendliness. While prior work flags beginner issues or predicts generic issue properties [7, 24, 29], we (i) identify *Candidates for University Projects* aligned with learning outcomes, (ii) operate at issue creation time using only title, body, and image descriptions, (iii) evaluate via a ranking-based Top-$K$ selector that matches a fixed review budget, and (iv) compare classical ML and LLMs under a blinded maintainer review. We also predict pedagogical descriptors (size, focus, and effort) to support instructor–task matching. Together, these choices bridge automated issue triage and classroom needs through an education-centered, early-signal, expert-in-the-loop approach.

## 3 Method

This section describes the research method we employed to identify OSS repository issues suitable for university students, referred to in this work as "Candidates for University Projects."

### 3.1 Our Case: JabRef Reference Manager

Our case study was conducted on JabRef, an actively maintained, OSS reference manager hosted on GitHub. JabRef is a mature OSS project with over 1.7 million downloads and a long history of academic use, making it particularly relevant for university settings [1]. It features an active contributor base, over 1,200 merged pull requests, and more than 300 active issues in the past year [25].

To support educational outreach and help onboard student contributors, the JabRef maintainers curate a dedicated list of issues known as the "Candidates for University Projects." This list is publicly available via a GitHub Project Board[1] and has been maintained since July 7, 2015. It highlights issues that the JabRef maintainers consider feasible for university students. The main curator of the Candidates for University Projects" list and a JabRef maintainer is also directly involved in computer science higher education. This dual role anchors curation in day-to-day teaching practice: items

---

[1]https://github.com/orgs/JabRef/projects/3

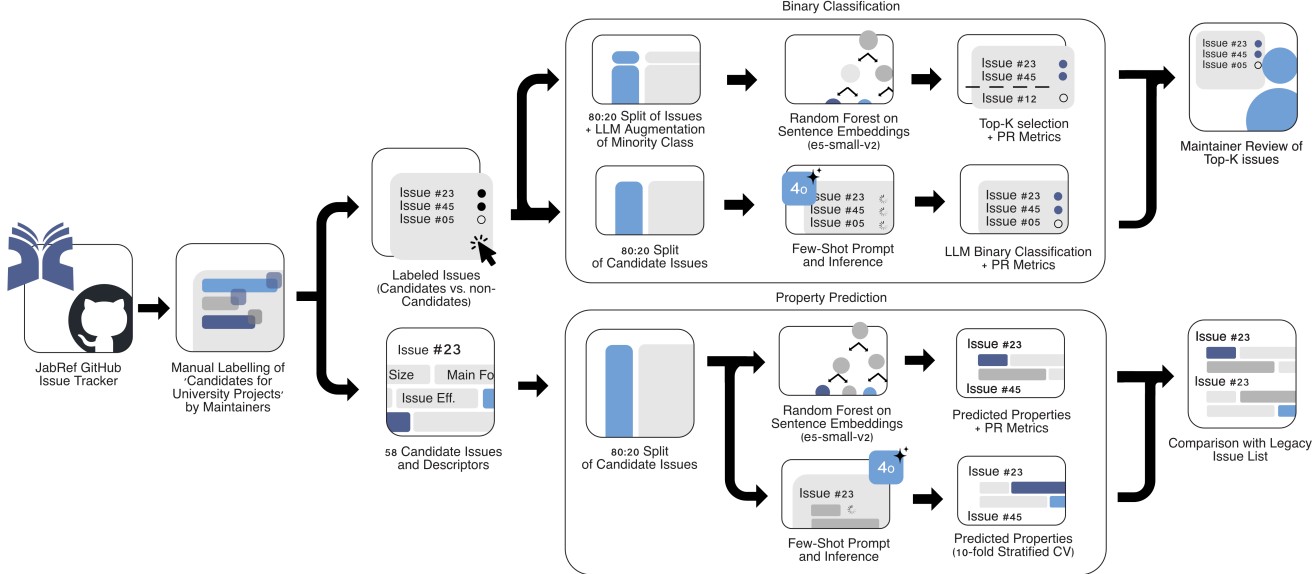

**Figure 1: Classification Methodology**

are selected for a clear problem statement, a bounded scope, and prerequisite alignment.

The "Candidates for University Projects" serves as a valuable starting point for instructors seeking real-world software tasks for coursework or capstone projects, and it reflects the maintainers' commitment to fostering meaningful educational participation in the JabRef community [13]. Each issue listed as a Candidate for University Projects" also includes a rich set of manually applied descriptive labels that capture key pedagogical and technical dimensions: Project Size, Issue Understanding Effort, Implementation Effort, Testing Effort, and Main Focus. For example, Issue Understanding Effort" reflects the background knowledge or contextual reading required to address a GitHub issue, while Implementation Effort" and "Testing Effort" capture the scope and intensity of actual coding and verification work. For instance, an issue might be labeled as small' in project size, low' in implementation effort, and focused on 'UI,' signaling it as a straightforward, student-friendly entry point for those interested in topics matching these characteristics.

### 3.2 Dataset Construction

We compiled two structured datasets from the JabRef issue tracker, each supporting a different task. The first dataset focused on identifying issues labeled by JabRef maintainers as Candidates for University Projects. To create this dataset, we extracted all issues from JabRef's GitHub repository created after July 7, 2015, the date when the "Candidates for University Projects" list was first established, and categorized each issue into two categories: issues included in the curated list and ones that were not. The snapshot of GitHub issue data was taken on April 21, 2025, ensuring that all issues analyzed were created before that date. This dataset yielded 3,295 total issues.

We restricted our feature set to only the issue's title, body, and image descriptions to simulate real-world applicability. This decision reflects the common scenario in which maintainers evaluate

an issue shortly after it is created, before additional information (e.g., labels, comments) has been added. Additionally, for issues that included attached images, we processed each image using the Moondream API (a lightweight, open-source vision–language model for fast image captioning and visual question answering) [17, 26] to extract AI-generated natural language descriptions. The generated captions were appended to the issue body text prior to embedding.

The second dataset targeted a multi-label classification task, including only issues manually classified as university projects by JabRef maintainers. From our binary classification dataset, we identified 53 issues that were in the curated "Candidates for University Projects" list. For each of these, we extracted the five manually applied descriptors: Size of Project, Issue Understanding Effort, Implementation Effort, Testing Effort, and Main Focus. Our classification task aims to preemptively infer these dimensions when the issue is first posted, based solely on its title, body, and image descriptions. As with the binary classification dataset, we appended Moondream-generated captions for any attached images, ensuring that visual context was also available to the multi-label prediction model.

We performed an 80/20 *stratified* split over the full dataset. We stratified our sample by splitting the 53 *Candidates for University Projects* issues into 42 for training and 11 for testing (approximately 80/20), and the remaining 3,242 issues not labeled as candidate issues were split to maintain the same class ratio in both folds. These two splits were then combined. This yielded a total of 2,636 training issues and 659 test issues, preserving class balance across splits. For the second dataset, used for multi-label classification, we restricted to only the 53 Candidate issues. These were split using the same 80/20 rule, resulting in 42 labeled examples for training and 11 for evaluation across the five target attributes. All Top-$K$ and classification metrics reported in this work are computed on the 659-issue binary test set and the 11-issue multi-label test subset.

We also explored and implemented a cosine similarity–based approach that estimates a target property for a new issue by comparing it to previously labeled university-appropriate examples. To compute similarity, we represent each issue (title, body, and image descriptions, including image description) with sentence-transformer embeddings (e.g., intfloat/e5-small-v2) and measure cosine similarity between the new issue's embedding and the embeddings of previously labeled university-appropriate tasks. The predicted class is then determined by majority vote from the Top-$K$ most similar labeled issues (i.e., those with the highest cosine similarity).

We used these datasets to classify the "Candidates for University Projects" issues and predict their properties. The following subsections describe the steps we followed to conduct the evaluation.

### 3.3    Random Forest Binary Classification

To predict whether a GitHub issue qualifies as a "Candidate for University Project," we develop a Random Forest binary classifier. Formally, this is a supervised learning problem with two possible outcomes: whether issues are predicted for university projects or not.

**Positive Class Augmentation.** A challenge emerged during dataset analysis: of the 3,295 tasks collected (restricted to those created after July 7, 2015), only 53 were labeled as such. This results in a university project representation of roughly 1.6%, which raised major concerns. To address the class imbalance, we employed an LLM-based positive-class augmentation step similar to the approach by Wysocki and Ochodek, who used LLMs to generate synthetic task descriptions to up-sample underrepresented categories.[28]. They showed that injecting hundreds of LLM-generated issues improved classification. Thus, for each maintainer-labeled positive issue, a GPT-4–class model generated multiple semantically faithful paraphrases of the title, body, and image descriptions. The augmented positives were appended to the original data, increasing the positive prevalence from 1.6% to 20% in our dataset. These 20% provide the learner with more signal without inventing new labels and degrading performance, according to the literature [5, 9].

**Sentence Embedding.** For each issue, the title, body, and image descriptions were concatenated and encoded into dense sentence embeddings using a sentence-transformer (e.g., intfloat/e5-small-v2). E5 embeddings are designed for general-purpose semantic representations and show strong performance across retrieval, clustering, and classification tasks [18, 27]. These embeddings allow the classifier to capture semantic similarity between differently worded issues (e.g., "documentation," "minor bug," "UI fix") and provide effective features for supervised models [27]. E5's contrastive training on weakly-supervised text pairs enables it to perform competitively with much larger models on the MTEB benchmark [27].

**Random Forest Prediction.** We then trained a Random Forest classifier on these embeddings. Random Forest is a robust baseline for tabular, dense representations and performed reliably in our setting, in line with prior evidence of their competitiveness for issue classification in OSS settings [21].

**Random Forest Evaluation.** We evaluated a ranking-based selector that returns the Top-$K$ issues, which avoids relying on calibrated probabilities in an imbalanced setting. We used $K \in \{5, 10, 20, 30\}$. This setup mirrors the practical use case of surfacing a fixed-size shortlist for instructors, while also enabling controlled exploration of the precision–recall trade-off by varying $K$.

### 3.4    LLM for Binary Classification

Complementing our Random Forest approach, we evaluated the potential of an LLM to perform the same ranking tasks. The goal was to assess whether a general-purpose, pretrained model could serve as an alternative for identifying and ranking issues by their suitability as "Candidate for University Projects" tasks. For this experiment, we used OpenAI's GPT-4o model, selected for its strong performance in few-shot classification and natural language understanding [31].

**Few-shot Learning.** We employed few-shot prompting because evidence from title–abstract screening for systematic reviews shows that one-shot and few-shot prompting outperform zero-shot prompting, and that GPT-4 with few-shot variants can achieve human-comparable screening performance, while redesigned zero-shot prompting still underperforms humans[11]. Each prompt included 10 representative examples from our training set (5 tagged university-appropriate issues and 5 not tagged as university-appro-priate issues), with each example containing issue title, description, and image descriptions, paired with its correct binary classification.

**LLM Prompting.** The prompt structure followed a standard few-shot template: it began with a task definition explaining what constitutes a university-appropriate issue, followed by 10 labeled examples and a batch of issues to classify and rank. Each issue's title, description, and metadata (including GitHub labels) were combined into a single input prompt, formatted consistently with the examples. We did not fine-tune or perform any additional training on the base model. The model's response was a ranked list of issues (best first), with the respective class they were classified as (appropriate "Candidate University Project" or not appropriate).

**LLM Output Evaluation.** We computed the Top-$K$ metrics at $K \in \{5, 10, 20, 30\}$, comparing the ranked issues against our ground truth labels and other baseline models. To assess robustness and reliability, we conducted a consistency experiment by running the ranking task three times, each with the same set of 10 selected examples as the few-shot input. This allowed us to evaluate the consistency of rankings across independent runs. We analyzed the Top-$K$ results ($K \in \{5, 10, 20, 30\}$) across all three runs to characterize overall performance and variance of the few-shot approach, comparing the overlap of Top-$K$ issues and rank stability across runs. The three runs showed high consistency: at $K = 30$, 29 of the top 30 issues appeared in all three runs, demonstrating the stability of the few-shot ranking approach despite minor variations in ordering.

### 3.5    Maintainer Review of Recommendations

To evaluate the practical utility of our classifier in supporting maintainers, we requested a JabRef maintainer to analyze the results of our recommendation. We compared the top-30 outputs from the Random Forest model and the LLM and chose the one with higher recall (i.e., more candidate-appropriate issues returned) for review.

To simulate an authentic early-triage scenario, the maintainer received only the textual content of each issue containing its title, body, and image descriptions. The maintainer did not have access to the model's confidence scores or the ranking. Still, we removed

any issue already present in JabRef's manually curated "Candidates for University Projects" to avoid confirmation bias. The remaining issues thus formed our evaluation set. We asked the maintainer to assess each issue's suitability as a "Candidate for University Project," following the same criteria they would use during JabRef's manual curation process. This procedure allowed us to examine whether the classifier could surface new, previously unrecognized candidate issues that align with expert judgment. Such validated issues could be incorporated into future training iterations, supporting a closed-loop improvement process in automated triage.

## 3.6 Property Prediction

For predicting properties of candidate issues (*Size of project*, *Issue understanding effort*, *Implementation effort*, *Testing effort*, and *Main focus*), we operate solely within the maintainer–labeled Candidate set ($n = 53$). Each issue's *title* and *description* are concatenated and encoded into a dense vector using a sentence transformer (i.e., intfloat/e5-small-v2).

Each descriptor is modeled as a single–label task with the following value sets: *Size of project* $\in$ {*small, medium, large*} (ordinal); *Issue understanding effort* $\in$ {*low, medium, high*} (ordinal); *Implementation effort* $\in$ {*low, medium, high*} (ordinal); *Testing effort* $\in$ {*low, medium, high*} (ordinal); and *Main focus* $\in$ {*UI, logic, logic + UI*} (nominal). For the four ordinal descriptors, we preserve the natural order (e.g., *low < medium < high*) when reporting results, while training uses standard multiclass classification. If an issue is tagged with both UI and Logic, it is assigned the logic + UI class. We trained separate Random Forest classifiers for each of the five descriptors, treating each as an independent multiclass classification task. For the baseline property prediction models reported in Tables 4-6, we used TF-IDF vectorization with 10-fold ShuffleSplit cross-validation. For comparison, we also trained models using sentence transformer embeddings (intfloat/e5-small-v2) with 400 estimators and balanced class weights, employing 5-fold stratified cross-validation during development to assess model performance and prevent overfitting given the small sample size of 53 labeled candidates.

During inference, each issue embedding is passed through all five trained models independently, with each model predicting the most likely label for its corresponding property. The separate models enable fine-grained predictions across multiple dimensions simultaneously, producing a complete characterization vector for each candidate issue.

We also explored the LLM's ability to infer the same descriptive labels used by maintainers to describe pedagogical and technical characteristics, namely, Size of Project, Issue Understanding Effort, Implementation Effort, Testing Effort, and Main Focus. Each of these was posed to the LLM in a few-shot format using tailored prompts, simulating how an AI assistant might preemptively tag issues for educational triage. Both prompts are available in the replication package [2]. In doing so, we aimed to understand whether an LLM could replicate or exceed the performance of traditional classifiers in this niche domain. Furthermore, the LLM's responses offered a form of reasoning that could be qualitatively analyzed, helping identify scenarios where the model identified contextually valid but previously unlabeled university-appropriate tasks.

## 4 Results

This section reports results for two tasks: (1) binary classification of issues as "Candidates for University Projects" and (2) prediction of descriptive tags. We compare a Random Forest model with GPT-4o. We use a ranking setup rather than a fixed threshold, returning the Top-K issues (K 5, 10, 20, 30) to match a realistic review shortlist and to study the precision–recall trade-off. The replication package is publicly available on Zenodo [2].

## 4.1 Binary Classification Performance

As described in the method, we adopt a ranking-based Top-$K$ selection strategy rather than applying a fixed probability threshold (with $K \in \{5, 10, 20, 30\}$). The resulting metrics at different $K$ are reported in Table 1 and detailed below

**Table 1: Performance Metrics at Different K Values**

| K | Precision | Recall | F1-Score | Accuracy | TP Found |
|---|-----------|--------|----------|----------|----------|
| 5 | 0.400 | 0.182 | 0.250 | 0.982 | 2/11 |
| 10 | 0.300 | 0.273 | 0.286 | 0.977 | 3/11 |
| 20 | 0.250 | 0.455 | 0.323 | 0.968 | 5/11 |
| 30 | 0.167 | 0.455 | 0.244 | 0.953 | 5/11 |

At $K = 5$, the model retrieves only 2 of 11 true candidates (precision **0.40**, recall **0.18**), meaning the shortlist misses most true positives despite higher precision. At $K = 10$, precision drops to **0.30** while recall improves slightly to **0.27** as one additional true candidate is recovered, but with more false positives. At $K = 20$, recall rises to **0.45** (the highest across all values) while precision declines to **0.25**, favoring coverage at the cost of more incorrect predictions. At $K = 30$, recall remains **0.45** and precision falls further to **0.17**, with no additional true candidates beyond those already retrieved at $K = 20$, indicating that most high-scoring positives have already been identified.

We also evaluated the effectiveness of an LLM for automatically identifying "Candidates for University Projects." We employed OpenAI's GPT-4o with a few-shot, zero fine-tuning classification setup. Our goal was to determine whether a general-purpose model could serve as a lightweight alternative to supervised learning for the binary classification task. The results of the output generated by the LLM are summarized in Table 2.

**Table 2: LLM Ranking Performance Metrics at Different K Values**

| K | Precision | Recall | F1-Score | Accuracy | TP Found |
|---|-----------|--------|----------|----------|----------|
| 5 | 0.200 | 0.091 | 0.125 | 0.979 | 1/11 |
| 10 | 0.100 | 0.091 | 0.095 | 0.971 | 1/11 |
| 20 | 0.050 | 0.091 | 0.065 | 0.956 | 1/11 |
| 30 | 0.067 | 0.182 | 0.098 | 0.944 | 2/11 |

At $K = 5$, the model retrieves only 1 of 11 true Candidates (precision **0.20**, recall **0.09**), so the shortlist has very low recall and surfaces just one true positive. At $K = 10$, no additional true Candidates are found, precision drops to **0.10**, and recall remains **0.09**. At $K = 20$, still no additional true Candidates are found, recall remains **0.09**, and precision declines to **0.05**. At $K = 30$, recall improves to **0.18** while precision is **0.07**, recovering one additional true Candidate compared to $K = 20$, but precision remains low.

The LLM ranking achieved low recall (0.091–0.182) across different K values, indicating limited ability to identify university-appropriate tasks within the top-ranked issues. However, precision remained low (0.050–0.200), reflecting that many highly-ranked issues were not actually labeled as University Projects by maintainers. At k=30, the model found 2 out of 11 candidates issues in the top-30 ranked issues, achieving a recall of 0.182. The three runs showed high consistency: at K=30, 29 issues appeared in all three runs, which demonstrates the stability of the few-shot ranking approach.

## 4.2 Expert Review of RF-Surfaced Issues

We provided the maintainer with the top-30 issues ranked by our Random Forest classifier, which we selected over the LLM based on its superior performance on our binary classification task. Our maintainer conducted a blind, independent review of these 25 new suggestions in a randomized order, judging whether they would consider each a viable candidate for student projects. The results were encouraging:

**Table 3: Performance Metrics at Different K Values**

| K | Legacy Cand. | New Cand. | Total Cand. | Precision |
|---|---|---|---|---|
| 5 | 2 | 3 | 5 | 1.000 |
| 10 | 3 | 5 | 8 | 0.800 |
| 20 | 5 | 9 | 14 | 0.700 |
| 30 | 5 | 13 | 18 | 0.600 |

13 of the 25 previously untagged issues were judged to be valid Candidates by the maintainer, mainly due to their clear descriptions, narrow scope, localized changes, and alignment with familiar technical concepts. The remaining 12 were not, typically due to unclear require- ments, high implementation complexity, or entanglement with fragile or legacy parts of the codebase. The model's top-ranked issues were most reliable: precision reached 100% at K=5 (all 5 issues accepted), then declined to 80% at K=10, 70% at K=20, and 60% at K=30 as lower- confidence predictions were included.

In trying to improve the classifier, we also analyzed the reasons for rejection for the other issues surfaced by the machine learning model. Of the 12 not accepted, the maintainer cited scope too large or too many components (4), unclear requirements or missing testable criteria (4), high technical effort (2), feature already exists (1), and complex legacy code (1); one issue contributed to both the scope and legacy counts, so reasons for rejection were not mutually exclusive.

## 4.3 Property Prediction

Building on the binary classification of university-appropriate tasks, we next evaluated whether specific pedagogical properties, such as implementation effort, testing effort, issue understanding effort, main focus, and size of project, could also be predicted. These attributes provide finer-grained support for aligning student contributors with appropriate tasks.

For each target property, we applied 5-fold cross-validation to ensure robust results and tested several feature configurations: using only the issue text (title and body), and augmented with image descriptions. These results appear in Table 4.

While the above model relied only on information available when an issue is posted, we also evaluated models that use features

**Table 4: Standard (title + body + image descriptions) Feature Results**

| Target | Acc | Prec | Rec | F1 |
|---|---|---|---|---|
| Implementation effort | 0.46 | 0.223 | 0.326 | 0.256 |
| Issue understanding effort | 0.45 | 0.295 | 0.340 | 0.304 |
| Labels | 0.15 | 0.013 | 0.075 | 0.022 |
| Main focus | 0.35 | 0.222 | 0.295 | 0.245 |
| Size of project | 0.50 | 0.245 | 0.340 | 0.276 |
| Testing effort | 0.38 | 0.211 | 0.291 | 0.243 |

added later, namely labels and comments. These features are not available for our real-time use case, but we include them for two reasons: (1) to measure their predictive value when they are available, and (2) to estimate an upper bound on performance when richer, community-curated information is present. Results for the label-only and comment-only variants appear in Table 5 and Table 6, respectively.

**Table 5: Label Feature Results**

| Target | Acc | Prec | Rec | F1 |
|---|---|---|---|---|
| Implementation effort | 0.48 | 0.216 | 0.332 | 0.258 |
| Issue understanding effort | 0.48 | 0.327 | 0.348 | 0.329 |
| Labels | - | - | - | - |
| Main focus | 0.42 | 0.277 | 0.334 | 0.295 |
| Size of project | 0.55 | 0.350 | 0.378 | 0.359 |
| Testing effort | 0.39 | 0.231 | 0.308 | 0.259 |

**Table 6: Comment Feature Results**

| Target | Acc | Prec | Rec | F1 |
|---|---|---|---|---|
| Implementation effort | 0.52 | 0.208 | 0.329 | 0.244 |
| Issue understanding effort | 0.45 | 0.148 | 0.287 | 0.192 |
| Labels | 0.12 | 0.025 | 0.075 | 0.035 |
| Main focus | 0.45 | 0.208 | 0.323 | 0.247 |
| Size of project | 0.55 | 0.166 | 0.300 | 0.211 |
| Testing effort | 0.38 | 0.114 | 0.255 | 0.152 |

While performance varies across feature configurations, additional textual signals, i.e. comments, can enhance our prediction quality. Labels performed best, but are not usable in our proposed automated pipeline, reinforcing the need for strong early signals and better lightweight metadata at issue creation time. As an interpretable, training-free alternative, we also evaluate a cosine-similarity retrieval baseline; results are summarized in Table 7.

**Table 7: Cosine Similarity Feature Results**

| Target | Acc | Prec | Rec | F1 |
|---|---|---|---|---|
| Implementation effort | 0.51 | 0.239 | 0.365 | 0.283 |
| Issue understanding effort | 0.49 | 0.305 | 0.369 | 0.330 |
| Labels | 0.14 | 0.013 | 0.069 | 0.021 |
| Main focus | 0.45 | 0.280 | 0.388 | 0.318 |
| Size of project | 0.59 | 0.385 | 0.419 | 0.395 |
| Testing effort | 0.43 | 0.262 | 0.337 | 0.291 |

As shown in Table 7, the cosine similarity delivered competitive results across several target properties. It achieved the highest F1-score for *Size of project* (0.395), and strong performance on *Issue understanding effort* (0.330), *Main focus* (0.318), and *Testing effort* (0.291). These results suggest that even without training a supervised model, semantic similarity to previously labeled issues can

guide educational descriptors prediction. This approach is particularly interesting in scenarios where labeled data may be limited or where interpretability is desired. Since the prediction is based on retrieving and comparing with known examples, instructors or maintainers can inspect the retrieved cases and understand the rationale behind the output. LLM-based label prediction results are summarized in Table 8.

**Table 8: LLM-Based Label Prediction Results**

| Target | Acc. | Prec. | Recall | F1-Score |
|---|---|---|---|---|
| Implementation effort | 0.300 | 0.206 | 0.500 | 0.288 |
| Issue understanding effort | 0.300 | 0.143 | 0.167 | 0.154 |
| Labels | 0.111 | 0.062 | 0.062 | 0.062 |
| Main focus | 0.400 | 0.212 | 0.438 | 0.267 |
| Size of project | 0.600 | 0.524 | 0.400 | 0.413 |
| Testing effort | 0.500 | 0.500 | 0.500 | 0.486 |

As shown in Table 8, GPT-4o was especially effective at predicting the "Testing effort" label, achieving an F1-score of 0.267 on the test set. Moderate performance was observed for "Size of project", "Main focus", and "Implementation effort" (F1-scores: 0.223, 0.178, 0.173). "Issue understanding effort" was more difficult for the model.

## 5 Discussion

This study addresses two central questions: (RQ1) To what extent can OSS issues be automatically classified as *Candidates for University Projects*? and (RQ2) To what extent can we predict pedagogically relevant properties of Candidate issues? Our results reveal that, even with extreme class imbalances, automatic classification is feasible and can successfully surface issues previously missed by maintainers. We also observed contrasts between Random Forest classifiers and GPT-4o: the former offered more conservative, stable predictions. Together, these insights highlight opportunities for automating the triage of educationally meaningful OSS issues and point toward hybrid strategies that combine the complementary strengths of classical models and LLMs.

### 5.1 Random Forest vs. LLM for Binary Classification

The Random Forest classifier, trained on sentence-transformer embeddings with positive-only augmentation, behaved conservatively in a ranking-based Top-$K$ evaluation. At $K = 5$, it achieved the highest precision (0.40) with a recall of 0.18, surfacing 2 of the 11 known Candidates. However, as $K$ increased, recall rose modestly to 0.45 at $K = 20$ and 30, but precision steadily declined, reaching 0.17 at $K = 30$. This indicates that the model reliably identifies a small core of "easy" positives early but struggles to find additional true Candidates beyond that. In contrast, the GPT-4o ranking baseline we evaluated achieved low recall (0.091–0.182) and low precision (0.050–0.200), finding only 1–2 out of 11 true candidate issues in the Top-$K$ ranked results. Overall, Random Forest significantly outperforms the LLM approach, providing both better ranking quality and more reliable identification of CUP issues. Random Forest provides a stable, higher-performing ranking, while the LLM's low recall and precision limit its practical utility for this task.

### 5.2 Classifier to Augment the Candidate List

Several issues initially assumed to be false positives when analyzing the Random Forest outputs (those not present in the original

"Candidates for University Projects" list were later assessed by the project maintainer as candidate tasks).

Given the sparsity of candidate issues in the full dataset (∼1.6%), a naive ranking would be expected to surface almost exclusively non-Candidate issues. That over half of the model's new suggestions passed expert review suggests that the classifier is surfacing real pedagogical value. These 13 accepted issues were not previously labeled by the maintainer, indicating that the model is not simply replicating existing judgment, but it is expanding and enhancing it.

Several issues initially assumed to be false positives when analyzing the Random Forest outputs (those not present in the original "Candidates for University Projects" list were later assessed by the project maintainer as candidate tasks. This outcome is significant on several fronts. Model-generated signal exceeds random or baseline selection. Given the sparsity of candidate issues in the full dataset (∼1.6%), a naive ranking would be expected to surface almost exclusively non-Candidate issues. That over half of the model's new suggestions passed expert review suggests that the classifier is surfacing real pedagogical value. The model complements human triage. These 13 accepted issues were not previously labeled by the maintainer, indicating that the model is not simply replicating existing judgment, but it is expanding and enhancing it. In time-constrained OSS projects, triaging hundreds of open issues for academic fit is difficult. A ranking system that can elevate strong candidates, even if imperfectly, reduces the cognitive and logistical burden on already overextended maintainers. The maintainer's feedback effectively "confirms" new data points that could be fed back into the training process. This positions the classifier not just as a one-off tool, but as a component of a semi-automated issue discovery pipeline, where model suggestions iteratively refine human-curated educational boards. Better than Good First Issues. Notably, several of the accepted candidates had never been tagged with GitHub's good first issue label. This underscores the need for more targeted, model-driven signals when surfacing university-appropriate tasks. Our classifier offers a novel alternative to community heuristics that tend to skew toward triviality.

### 5.3 Implications for Practice

Several practical steps for both educational tooling and OSS maintenance are revealed by the evaluation of our results and findings. First, that a rank-then-review workflow is viable: presenting instructors a small Top-$K$ (e.g., 5–30) shortlist each cycle meaningfully reduces triage effort while keeping attention on the highest-scoring issues. Second, repositories can improve predictability by enriching early issue text (clear problem statements, acceptance criteria, and scope cues), which directly benefits embedding-based models. Third, Random Forest provides a reliable ranking approach that outperforms LLM-based methods, making it the preferred choice for identifying university-appropriate issues without requiring additional LLM passes. Finally, incorporating maintainer feedback on surfaced candidates closes the loop, enabling active learning and steadily improving Recall@K without increasing reviewer workload.

### 5.4 Future work

Rather than making definitive classification decisions, future work would see us developing an assistant integrated directly into GitHub

that flags potentially student-appropriate issues in real time as they are posted. This tool would act as a triage support system, automatically surfacing issues that are likely suitable for the Candidates for University Projects list and suggesting their properties (e.g., implementation effort, issue understanding effort, testing effort, and main focus).

The system's primary role would be to reduce triage burden, not replace human judgment. Proactively identifying promising issues can prevent high-value tasks from being missed due to time constraints or high issue volume. This system could be implemented as a GitHub bot (e.g., using Probot) that activates upon new issue creation or in response to a maintainer's manual trigger (such as @JabBOT classify). In addition to this, the bot would process issue titles, descriptions, and images, generate predictions using our trained models, and post a structured comment or label indicating the issue's likely suitability and educational profile. In the long term, the assistant could evolve into a recommendation service that helps match university students with OSS tasks aligned to their skills and academic goals. Maintainers could also provide feedback to refine the model over time, creating a feedback loop that strengthens both model accuracy and community trust.

## 6 Limitations

This work is constrained by limitations that affect the scope and interpretation of our results. These limitations should be considered when interpreting the results and building on current work in the future. Foremost, the dataset is small; the Candidates for University Projects dataset contains only 53 labeled examples, which restricts the statistical power of our experiments and limits the generalizability of the supervised models to other OSS projects or domains. Although these 53 examples are real maintainer judgment, they still weaken the models and limit how well they transfer to other projects and how well they perform in edge cases.

Furthermore, although comments, labels, and other post-creation metadata were shown to improve prediction quality in exploratory analyses, such information is frequently unavailable at the moment of issue creation, the stage at which early triage is most valuable. This means our models operate with an intentionally constrained feature set, potentially underestimating the achievable performance in scenarios where richer contextual signals are available.

Similarly, while incorporating image-based features provided additional value for certain issues, the absence of images in a substantial portion of the dataset reduces the consistency and applicability of these features in practice. The uneven distribution of these inputs may also bias model learning toward text-only patterns. Additionally, we acknowledge the potential for pretraining leakage. Since JabRef is a public repository and the "Candidates for University Projects" list has been available online, it is plausible that the sentence-transformer model (i.e., intfloat/e5-small-v2) used in this work may have encountered issues labeled as candidates during its large-scale web-based pretraining. While we mitigate this risk by using stratified train/test splits and not providing model access to the list, the possibility of representation-level leakage cannot be fully ruled out.

Finally, our evaluation was conducted exclusively within the context of the JabRef project. While JabRef represents a mature OSS repository with an active maintainer community, its domain, contributor base, and labeling practices may not fully represent those of other projects. Future studies should examine the transferability of these methods to diverse OSS ecosystems, larger datasets, and multilingual contexts to validate their broader utility.

## 7 Conclusion

This study compares traditional machine learning and LLMs to support early triage of open-source software issues for academic engagement. By focusing on identifying "Candidates for University Projects" and predicting relevant properties, we show that automated methods can complement the manual curation currently required from OSS maintainers and educators.

Our study highlight that, even with an extreme class imbalance and limited labeled data, textual features combined with a ranking-based selection strategy can effectively identify issues. With maintainer validation confirming that a majority of top-ranked predictions were indeed suitable for student work, we demonstrate that models can surface candidate tasks that were previously untagged. Random Forest models demonstrate strong performance in this task, reliably identifying suitable issues and offering a practical foundation for downstream filtering. In contrast, LLM-based approaches showed limited effectiveness, achieving low recall and low precision, which limits their practical utility for this specific task. The proposed GitHub-integrated tool represents a natural extension of this work, offering real-time, minimally intrusive classification within existing OSS workflows. Beyond improving efficiency for maintainers, such automation could strengthen the pipeline between OSS communities and academic programs, enabling students to engage with well-scoped tasks from the moment they are created. Ultimately, this research contributes toward building a scalable, data-driven infrastructure that not only reduces triage burdens but also fosters sustainable collaboration between educational institutions and the open source ecosystem.

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
