# OpenReview forum: "Identifying “Candidates for University Projects” in Open Source Software"
_ACM.org/AIWare/2026/Conference — Submitted to AIware 2026_

### Official Review · Reviewer_WKVS · 2026-02-24

**Rating:** 1
**Confidence:** 5

**Review:**

(+) Strength:

+ The study addresses an interesting topic, especially for educators who would like to integrate OSS into coursework.

+ The idea of supporting issue triage with automated tools is aligned with ongoing research on OSS onboarding and education.

(-) Weaknesses:

- The study evaluation is limited in scope, as it is centered on a single repository (JabRef), which significantly affects external validity.

- The study dataset is small and highly imbalanced, and there is insufficient detail regarding dataset construction, annotation process, and labeling reliability.

- The binary classifier is trained on only 53 real positive examples, which is critically small for supervised learning and increases the risk of overfitting.

(I) Detailed Comment:

- It is not clear how “Candidates for University Projects” differ from commonly used labels such as “Good First Issue (GFI).” A discussion clarifying similarities and differences would strengthen the conceptual contribution.

- The authors provided a replication package that supports transparency.

- The evaluation is limited in scope, being centered on a single repository, JabRef, which affects the study's external validity.

- The binary classifier is trained on only 53 real positive examples. This is critically small for supervised learning, especially when using high-dimensional embeddings, which also makes the model highly susceptible to overfitting.

- The performance reporting is weak and potentially misleading. Identifying 5 out of 11 known candidates (≈45% recall on a tiny subset) is not statistically convincing.

- The precision calculation relies on subjective maintainer judgment without a clear protocol, raising concerns about bias and reproducibility.

- The LLM baseline is under-specified. It is unclear how it was prompted, fine-tuned, or configured, which could make the comparison unfair.

- To improve readability, consider including a summary box that synthesizes the main findings related to each RQ.

- Minor:

- Keep the acronyms use consistent (e.g., LLM, ML) and avoid redefining them multiple times.

- Figure 1 is not referenced in the main text.

- Fix formatting: “categories.[28].”

- Fix word “require- ments.”

**Summary:**

The study aims to automate the identification of OSS issues suitable for university coursework by comparing a supervised ML classifier trained with a random forest to an LLM baseline trained on labeled GitHub issues to flag Candidate University Projects.

---

> ### Author Response · Authors · 2026-03-18
>
> We thank the reviewer for the thorough and constructive feedback. We address each point below.
>
> **Distinction between Candidates for University Projects and Good First Issues.** We agree this conceptual distinction deserves sharper emphasis earlier and will consolidate it into a concise, dedicated paragraph in the introduction.
> However, we would like to highlight that this distinction is already drawn in multiple places in the paper (Introduction, Background, Section 3.1, and Section 5.2, where we note explicitly that GFI labels are "typically scoped for one-time, low-effort contributions, instead of meaningful opportunities for academic projects," and that several of our accepted candidates had never been tagged as GFIs at all.
>
> **Single-repository scope and external validity.** JabRef is not an arbitrary choice. It is, to our knowledge, the only OSS project that maintains a publicly curated, maintainer-validated list of issues explicitly designated for university student engagement, making it the only available ground truth for this task. This is a deliberate tradeoff: by grounding the study in a single, well-understood repository with a consistent, expert-validated labeling process, we gain strong internal validity (we know exactly what the labels mean, who produced them, and under what criteria). Expanding to multiple repositories at this stage would introduce heterogeneous and incompatible labeling practices, undermining the very ground truth the classifier depends on. External validity is appropriately sacrificed in favor of methodological rigor at this early stage of the research, consistent with standard case study methodology in software engineering research. We acknowledge this explicitly in Section 6 and position the work as a proof-of-concept designed to motivate future cross-project generalization, ideally as other OSS communities adopt similar curation practices, enabled in part by tools like the one we propose.
>
> **Dataset size and overfitting risk.** We agree that 53 positive examples is a small sample (and we say so explicitly in the limitations section). However, several design choices directly mitigate overfitting risk: positive-class augmentation brings the effective training prevalence from 1.6% to 20%; sentence-transformer embeddings (intfloat/e5-small-v2) provide pretrained, generalizable semantic representations rather than task-specific features; and a stratified 80/20 split preserves class balance across folds.
> The real-world validation by the maintainer (who confirmed 13 of 25 previously untagged issues as genuine candidates) provides clear evidence that the model is capturing meaningful signal, not memorizing training examples. The 53 examples are not a flaw in our study design; they are a faithful reflection of how sparse such lists are in practice, which is the problem motivating this work.
>
>
> **Performance reporting.** The reviewer characterizes identifying 5 of 11 known candidates as unconvincing. We would reframe this: the more meaningful result is that when the maintainer reviewed the full top-30 shortlist, 18 of 30 issues were confirmed as valid candidates (60% precision), including 13 that had not been tagged before. Given a baseline positive rate of 1.6% in the full dataset, a naive ranker would be expected to surface almost no true positives. The model's ability to concentrate genuine candidates in a short reviewable list is the practical contribution, and maintainer validation provides a meaningful signal beyond the labeled test set.
>
> **Maintainer judgment and bias.** The evaluator was the primary curator of the "Candidates for University Projects" list since its inception in 2015. The same person whose prior judgments constitute the ground truth labels for the entire study. Their assessment of new issues, therefore, reflects the same criteria applied consistently. To minimize bias further, we removed all previously tagged issues from the review set, stripped confidence scores and rankings, and randomized issue order before presenting them. We will describe this more explicitly in the paper.
>
>
> **LLM baseline specification.** Full prompt templates are available in the replication package. We will add a pointer to the replication package where we mention the prompt templates in the paper and explicitly point to them in the repository README to lower the barrier for finding the information.  We will also add temperature, model version, and other configuration details in the body of the paper.
>
>
> **On minor comments**. We will add a findings summary box per RQ, enforce consistent acronym usage, add an explicit Figure 1 reference in the main text, and fix the identified formatting errors.

---

> > ### Comment · Reviewer_WKVS · 2026-03-19
> >
> > Dear Authors,
> >
> > Thank you for the detailed and thoughtful response.
> >
> > I appreciate the clarifications provided. However, while I understand and accept the tradeoff in using a single repository to ensure strong internal validity, this choice significantly constrains the scope of the contribution. As it stands, the study remains a proof-of-concept case study, and the current evaluation does not provide sufficient evidence that the approach generalizes beyond this specific context. I would expect either broader empirical validation or a stronger positioning explicitly framing the work as an exploratory step with limited claims.
> >
> > Additionally, despite the mitigation strategies described, the very small number of positive examples (n=53) continues to raise concerns about model robustness and potential overfitting. Overall, while I see value in the problem and the direction of the work, I believe the current manuscript does not yet provide sufficient empirical strength and rigor to support its claims at this stage. I encourage the authors to build on this foundation by strengthening the evaluation and clarifying the contribution's scope and positioning.

---

> > > ### Author Response · Authors · 2026-03-19
> > >
> > > Thank you for the quick response.
> > >
> > > Please, allow us to take a step back and clarify the framing of this work's contribution. The idea of evaluating a generalizable classifier that could transfer across arbitrary OSS repositories is currently unachievable. To the best of our knowledge, no other OSS project maintains a publicly curated, maintainer-validated set of issues explicitly designated for university-level academic engagement at the scale required for supervised learning. JabRef's "Candidates for University Projects" list is unique in that sense. This is why we characterize our work as a proof-of-concept case study, a common design in software engineering research for demonstrating feasibility before broader generalization is possible, with several papers published in major venues following this approach. We expect that our paper will encourage other OSS communities to adopt similar curation practices, benefiting many students and making cross-project studies tractable in the future. If the reviewer is aware of other OSS projects that maintain comparable curated lists of university-appropriate issues, we would welcome those suggestions and be happy to incorporate them.
> > >
> > > Regarding the small number of positive examples (n=53), we agree this is a limitation and state so explicitly. However, this is not a flaw in study design; it is the reality of the problem domain. As discussed in the previous paragraph, this dataset is unique, and JabRef offers many more issues tagged as candidates for university projects than other OSS projects. A classifier that achieves 60% precision at k=30, with a background positive rate of 1.6%, and confirms 13 previously untagged issues as valid candidates, demonstrates meaningful signal under the specific conditions that make this problem hard. The class imbalance itself is not incidental; it is the core challenge motivating automated triage. Showing that a model can perform under these constraints is part of the contribution, not a prerequisite to be solved before the study begins.

---

### Official Review · Reviewer_wjiZ · 2026-03-08

**Rating:** 1
**Confidence:** 4

**Review:**

The problem addressed by the paper is potentially valuable: identifying well-scoped open-source issues for educational use could help instructors integrate OSS projects into coursework and reduce triage effort for maintainers. However, the current version of the paper suffers from several conceptual, methodological, and reporting weaknesses that limit the validity and reproducibility of the results.

**Strengths**

The paper tackles an important and practical problem at the intersection of software engineering education and open-source collaboration. If executed rigorously, the approach could help build datasets of educationally suitable issues and potentially support future work on evaluating AI-based coding agents. The use of maintainer feedback to assess surfaced issues is also a promising evaluation direction.

**Weaknesses**

Problem framing and clarity.
The abstract and introduction do not clearly articulate the precise problem being solved or why it matters. In particular, the notion of a “valuable issue” or “Candidate for University Projects” is not clearly defined early in the paper. The explanation of this concept appears much later in the manuscript, which makes it difficult for readers to understand the motivation and scope of the work.

Methodological concerns and fairness of comparison.
The comparison between Random Forest and the LLM baseline is problematic. The Random Forest model is trained on embeddings derived from a transformer-based model itself, effectively leveraging contextual representations, while the LLM baseline appears to operate only on the raw textual prompt. As a result, the Random Forest model is provided with richer contextual information than the LLM. For a fair comparison, the LLM should receive equivalent context (e.g., embedding-derived summaries). Furthermore, the Random Forest pipeline itself relies on transformer embeddings, which makes the claim that it “outperforms a transformer-based approach” somewhat misleading and circular.

LLM usage and reproducibility issues.
Several critical details necessary for reproducibility are missing. The paper does not report key LLM configuration parameters such as temperature, seed, or other generation hyperparameters. Additionally, the paper uses an LLM to generate synthetic positive examples for upsampling but does not provide any manual validation of these generated samples. Without evaluating the quality of the generated data, it is unclear how much noise this step introduces into the training data and how it affects the results.

Limited evaluation of LLM approaches.
The study evaluates only a single proprietary LLM. Given that the paper’s conclusions emphasize the relative weakness of LLMs compared to classical models, it would be more appropriate to evaluate multiple LLMs, including at least one open-source model. This would improve the robustness of the comparison and allow other researchers to reproduce the experiments.

Research question alignment and implications.
RQ2 (predicting contextual characteristics of candidate issues) appears largely disconnected from the central goal of identifying suitable university projects. The outputs of RQ2 could have been used to improve the primary classification task or refine candidate selection, but this connection is not explored. Additionally, the implications of the results for the stated research questions are not clearly articulated within the RQs.

Ambiguity around maintainer involvement.
The paper mentions that a maintainer validated the results but does not clearly explain who the maintainer is, how many maintainers were involved, or what level of teaching or mentoring experience they have with student projects. This information is important because the study relies heavily on the maintainer’s judgment as ground truth.

Presentation and writing issues.
The paper contains a few formatting and editing problems, including stray closing quotation marks and random dashes appearing between words throughout the manuscript.

Overall Assessment

The paper addresses an interesting and potentially impactful problem, and the idea of automating the discovery of educationally suitable OSS tasks is promising. However, the current study has several significant shortcomings, including unclear problem framing, methodological inconsistencies in the model comparison, missing reproducibility details, limited evaluation of LLM approaches, and weak alignment between research questions and the paper’s main objective.

For these reasons, I recommend rejection in its current form. A substantially revised version that clarifies the problem definition, strengthens the experimental design, improves reproducibility, and better aligns the research questions with the main objective could potentially make a stronger contribution.

**Summary:**

This paper studies the problem of automatically identifying open-source issues that could serve as suitable university-level software engineering projects. Using the JabRef repository as a case study, the authors construct a dataset of GitHub issues labeled as “Candidates for University Projects” by maintainers and train a Random Forest classifier on sentence-transformer embeddings to identify such issues. The study also evaluates a few-shot LLM baseline (GPT-4o) for ranking candidate issues and performs a maintainer validation of the model’s top-ranked suggestions. Additionally, the paper attempts to predict several pedagogical attributes (e.g., implementation effort, testing effort, and project size) for candidate issues using both classical machine learning and LLMs.

---

> ### Author Response · Authors · 2026-03-18
>
> We thank the reviewer for the thorough and constructive feedback. We address each point below.
>
> **Problem framing and clarity.** We note that the motivation for the work is established in the opening paragraphs of the introduction, which describe the burden on maintainers and the difficulty students face in finding appropriately scoped tasks. Section 3.1 then provides a detailed definition, including the five pedagogical descriptors and the curation criteria used by JabRef maintainers. That said, we agree the concept could be defined explicitly earlier in the paper, and in the revision, we will add a concise definition in the introduction so readers do not need to read ahead to scope the classification task.
>
>
> **Fairness of comparison.** We appreciate this observation and want to clarify the intent of the comparison. The Random Forest pipeline uses fixed, pretrained sentence-transformer embeddings as features. It does not fine-tune or adapt the transformer in any way. The LLM baseline, as you said, operates with full access to raw issue text via few-shot prompting, which is its natural and intended mode of use.
> However, it was never the goal to claim that a "non-transformer approach outperforms a transformer-based approach." Both pipelines draw on transformer representations at some level. We will revise the framing to make this distinction precise and remove any wording that implies a broader architectural conclusion. The comparison remains meaningful as a practical one: given the same input signal and no task-specific fine-tuning, which approach better surfaces university-appropriate issues?
>
>
> **LLM configuration and reproducibility.** The replication package contains the full prompt templates and scripts used in the paper. We agree, however, that key configuration parameters (temperature, model version, and API settings) should appear in the paper. We will add a dedicated reporting paragraph in the revision.
>
> Regarding the augmented positives, we note that LLM-based upsampling for class-imbalanced datasets is an established technique (Wysocki & Ochodek, 2024, cited in the paper), and the augmentation step generates semantically paraphrases of existing maintainer-labeled issues rather than inventing new ones. This bounds the noise risk. We will add a transparency note on this and include sample augmented issues in the replication package so readers can assess their quality directly
>
> **Single LLM evaluation.** The GPT-4o baseline was intended as a practical reference point (since it was accessible, widely used in software engineering research, and strong at few-shot tasks) rather than a comprehensive benchmark of LLM capabilities. We will clarify this positioning explicitly so the conclusion is read as "a standard few-shot GPT-4o baseline underperforms a supervised approach on this task" rather than "LLMs in general cannot do this." Broader evaluation across multiple models is our current step, and we will frame it as such in the future work section.
>
> **RQ2 alignment.** RQ2 is intentionally complementary to RQ1 rather than a refinement of it. Predicting pedagogical properties (effort, scope, focus) serves a distinct and practical purpose: once a candidate issue is surfaced, instructors need fine-grained descriptors to match it to students with appropriate skill levels and time constraints. This two-stage design mirrors how triage actually works in practice. We will make this pipeline logic explicit in the introduction and discussion so the connection between the two RQs is clear.
>
> **Maintainer involvement.** The maintainer who reviewed our recommendations is the primary curator of JabRef's "Candidates for University Projects" list, has maintained it since its inception in 2015, and has university teaching experience in software engineering, which directly informs their curation criteria. This dual role (OSS maintainer and CS educator) is precisely what makes their judgment credible ground truth for this task. We will describe this background more explicitly in the paper.
>
>
> **Presentation issues.** We will correct all formatting issues, including stray quotation marks and errant dashes, in the revision.

---

### Official Review · Reviewer_nKKY · 2026-03-11

**Rating:** 2
**Confidence:** 3

**Review:**

Strengths:

+This research contributes to building a scalable, data-driven infrastructure that both reduces triage burdens and supports sustainable collaboration between educational institutions and the open-source ecosystem.

+The replication package is publicly available on Zenodo. This fosters the reproducibility of the reported study and increases its transparency.

+The proposed approach seems novel and relevant for the AIWare community.

Weaknesses:

-The authors claimed that each of their prompts included ten representative examples from their training set, but they did not explain how they specifically chose these examples, nor did they investigate the impact of choosing specific examples over others on the quality of the results.

-The authors did not provide a template of the prompts they used in the study. This hinders the reproducibility of the proposed work.

-Although there is a replication package, the configuration (e.g., temperature) of the LLM used in the experiments is not sufficiently discussed in the paper. The authors should refer to Section IV.C of the paper by Wagner et al. (see reference below) to know how to better describe such a configuration.
Wagner, S., Barón, M. M., Falessi, D., & Baltes, S. (2025, May). Towards evaluation guidelines for empirical studies involving llms. In 2025 IEEE/ACM International Workshop on Methodological Issues with Empirical Studies in Software Engineering (WSESE) (pp. 24-27). IEEE.

-The LLM used in the study is known to be non-deterministic, so its outputs may vary from one execution to another when prompted with the same input. However, the authors never discussed this non-determinism in their work and did not explain how they tried to address it, and did not report the variance and the average of their results, etc.

-The authors concluded that their LLM-based approach showed limited effectiveness, achieving low recall and low precision, which limits its practical utility when it comes to the early triage of open-source software issues for academic engagement. However, the authors never tried to combine few-shot with other prompting strategies such as Chain-of-Thought, whereas such combinations are usually very efficient. Finally, the authors only experimented with a single LLM, whereas other LLMs such as Claude are known to be extremely efficient when completing various software engineering tasks.

Minor comments:

-The authors claimed that they employed few-shot prompting because evidence from title–abstract screening for systematic reviews shows that one-shot and few-shot prompting outperform zero-shot prompting. But they did not specify the references of these reviews.
-Figure 1 is blurred and therefore hard to decipher.

**Summary:**

This paper reports a study that compares a traditional machine learning approach and a large language model (LLM)-based approach for the early triage of open-source software issues, focusing on identifying tasks suitable for university student projects. Results show that automated methods can support the manual curation typically performed by maintainers and educators. They also show that the traditional ML-enabled approach supported by Random Forest models yields strong performance in this task, while the LLM-based approach exhibits limited effectiveness.

---

> ### Author Response · Authors · 2026-03-18
>
> We thank the reviewer for their careful reading and constructive feedback. We address each point below.
>
> **Selection of the few-shot examples.** The ten examples were selected at random from the training set (5 positive, 5 negative). We will make this explicit in the paper. We agree that investigating the sensitivity of results to example selection would be a valuable addition; we will include this as a direction for future work. That said, we ran the LLM ranking three times with the same fixed examples and observed that 29 of the top-30 issues appeared across all three runs. This provides evidence that the results are stable under the current selection strategy.
>
> **Prompt templates and reproducibility.** The prompt templates for both the binary classification and property prediction tasks are already available in the replication package (PropertyRF/scripts/predict_properties_llm.py and Binary/scripts/rank_issues_with_gpt4o.py). We will add a pointer to the replication package where we mention the prompt templates in the paper and explicitly point to them in the repository README to lower the barrier for finding the information.
>
> **LLM configuration reporting.** We agree that the LLM configuration is underreported. We will expand the description of the experimental setup in the revision, including temperature, API version, and other relevant hyperparameters, following the reporting guidance proposed by Wagner et al. (2025).
>
> **LLM non-determinism.** We are aware of this issue and partially addressed it by running the LLM ranking three times and measuring output consistency. However, we agree that this is not sufficiently foregrounded in the paper. We will add non-determinism as a limitation, clarify our mitigation strategy, and report run-to-run variance more explicitly so that readers can appropriately calibrate their interpretation of the LLM results.
>
> **Alternative prompting strategies and LLMs.** Great! Exploring Chain-of-Thought prompting or other LLMs would be a natural extension. However, the purpose of the LLM baseline in this paper was to establish a simple, accessible, zero-fine-tuning reference point, not to determine the full capability ceiling of LLMs on this task. Systematic exploration of alternative prompting strategies and models (including Claude and others noted for strong SE task performance) is explicitly part of our current work. Importantly, the central contribution of this paper is demonstrating that a lightweight, supervised model trained on a small, highly imbalanced dataset can surface genuine candidate issues at a level that withstands expert review. A result that holds regardless of where the LLM ceiling ultimately lies.
>
> **On minor comments.** We will add the references supporting our choice of few-shot over zero-shot prompting (Brown et al., 2020; Huotala et al., 2024, both already cited in the paper). We will also replace Figure 1 with a higher-resolution version in the revision